# Combining GAN with reverse correlation to construct personalized facial expressions

**Sen Yan**[1]*, **Catherine Soladié**[1], **Jean-Julien Aucouturier**[2], **Renaud Seguier**[1]

1 CentraleSupelec, IETR, Rennes, France, 2 Institut FEMTO-ST, Université de Franche-Comté, SUPMICROTECH, CNRS, Besançon, France

* yansen0508@gmail.com

## Abstract

Recent deep-learning techniques have made it possible to manipulate facial expressions in digital photographs or videos, however, these techniques still lack fine and personalized ways to control their creation. Moreover, current technologies are highly dependent on large labeled databases, which limits the range and complexity of expressions that can be modeled. Thus, these technologies cannot deal with non-basic emotions. In this paper, we propose a novel interdisciplinary approach combining the Generative Adversarial Network (GAN) with a technique inspired by cognitive sciences, psychophysical reverse correlation. Reverse correlation is a data-driven method able to extract an observer's 'mental representation' of what a given facial expression should look like. Our approach can generate 1) personalized facial expression prototypes, 2) of basic emotions, and non-basic emotions that are not available in existing databases, and 3) without the need for expertise. Personalized prototypes obtained with reverse correlation can then be applied to manipulate facial expressions. In addition, our system challenges the universality of facial expression prototypes by proposing the concepts of dominant and complementary action units to describe facial expression prototypes. The evaluations we conducted on a limited number of emotions validate the effectiveness of our proposed method. The code is available at https://github.com/yansen0508/Mental-Deep-Reverse-Engineering.

## 1 Introduction

Facial expression manipulation (FEM) is an image-to-image translation technique that aims to automatically edit face photographs of real humans to change their appearance [1, 2]. With the recent development of deep learning methods [3–6], FEM techniques have become highly realistic and attracted increasing attention in the media and general public, such as the ubiquitous face filters in TikTok, Instagram, and Zoom and the computer graphic animations in movies and video games.

However, most FEM techniques have three weaknesses. **Lack of fine control**. For most FEM-based systems, the smallest editable components of manipulation are the global attributes, such as emotion labels [4, 5]. Although these systems can synthesize different types of facial expressions, the facial expression prototype of each emotion is often unique. What if an AI system is required to generate a happy face only with "smiling" eyes while the other areas of

**Funding:** Our work was funded by Randstad group (https://www.randstad.fr/, to Sen Yan, Catherine Soladié, and Renaud Seguier), Agence Nationale de

la Recherche (ANR) REFLETS (https://anr.fr/, to Catherine Soladié, Jean-Julien Aucouturier, and Renaud Seguier), and European Research Council (ERC) PoC ACTIVATE 875212 and Fondation Pour l'Audition FPA RD-2018-2 (https://erc.europa.eu/homepage, to Jean-Julien Aucouturier). The funders had no role in study design, data collection and analysis, decision to publish, or preparation of the manuscript.

**Competing interests:** The authors have declared that no competing interests exist.

the face remain neutral? These FEM techniques are unable to handle such tasks. **Lack of ability to be personalized**. Most FEM-based systems are based on the emotional prototypes defined by Ekman et al., which are supposed to be universally perceived by humans [7, 8]. However, the universality of Ekman's prototypes is now being challenged by a growing number of psychologists [9–11]. Indeed, facial expression prototypes should be diverse among different people. Yet, there is no FEM-based application that can generate their personal facial expressions. **Lack of variety**. As research in psychology covers, there are more than 4000 labels of emotions [12]. Due to the limitation of large and reliable labeled data for training, most AI tools can only deal with Ekman's basic emotions, i.e., happiness, anger, sadness, surprise, fear, and disgust [13]. Non-basic emotion labels, such as self-confidence, are not explicitly available in existing databases. In addition to the lack of large labeled data, creating such a database with various emotion labels comes with many concerns: 1) time-consuming for the annotation and 2) requiring expertise (e.g., certified FACS coders [13]) for some labeling tasks.

## 1.1 Requirements

To address more critical domains, such as psychotherapy or the service industry, AI applications should describe, understand or detect more emotions in real life. For instance, one can imagine training to express self-confidence before a job interview or dealing with anxiety in a therapeutic context [14]. Thus the FEM-based process needs to adapt to more various and fine-grained requirements.

- **Flexibility**. The process should be capable of personalizing facial expressions for observers. That is, generating the desired expression that meets the need of the observer. As in most literature, we refer to the person whose face is being manipulated as an "actor", and to the person supervising/designing the manipulation as an "observer". In case they want to personalize their own facial appearance, the actor and the observer can be the same person.

- **Exhaustiveness**. The process should be applicable to any expressions, and not limited to basic facial expressions. This can be complex emotions or social attitudes (e.g., self-confidence) as well as more general expressions (e.g., how do you want to be seen during your job interview?).

- **Expertise-free**. The process should be controllable by any observers in a precise and consistent manner without the need for expert knowledge (e.g., FACS-certified coders, and knowledge in affective computing).

## 1.2 Contribution

In this paper, we propose a novel interdisciplinary approach (see Fig 1) to personalize facial expressions by combining Generative Adversarial Networks (GANs) [15] with a technique inspired by the cognitive sciences, psychophysical reverse correlation [16, 17]. Reverse correlation is both an experimental procedure and an analysis technique able to extract facial prototypes, or 'mental representations' of any given desired facial expression from the observer. In other terms, our approach identifies which attributes need to be modified to better fulfill the need of the observer, resulting in the generation of personalized facial expressions. This can meet the requirement of **Flexibility**.

Differing from typical GANs that can manipulate facial expressions, our approach has the following strengths.

- **Exhaustiveness**. One can address any emotion or social attitude, with no need to build a dedicated training database for each emotion or social attitude. Rather, we use local attribute

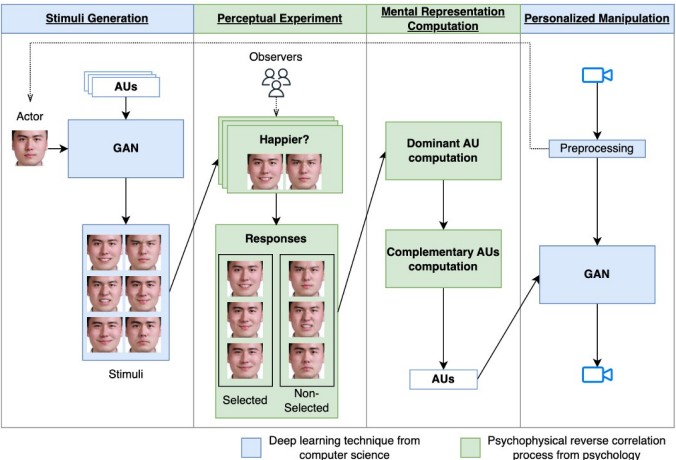

**Fig 1. The framework of our approach to personalize facial expressions.** We combine the recent deep learning technique, i.e., Generative Adversarial Network (highlighted in blue), with psychophysical reverse correlation, a recently emerging technique from the cognitive sciences (highlighted in green). We employ the same GAN to extract personalized control parameters (i.e., mental representation) and to personalize facial expressions.

to manipulate facial expressions, which can cover a wide range of local facial movements, in order to reproduce the observer's prototype, regardless of what the prototype's target expression is.

- **Expertise-free**. No expert knowledge in affective computing or certified FACS coder [13]) is needed to create the personalized prototype since our approach only requires the observer's perception (i.e., subjective judgment) rather than the observer's expertise.

Conversely, differing from typical reverse correlation approaches which use 3D virtual avatars [18], using FEM techniques (such as GANs) allows manipulating real faces (2D pictures), thus providing an easier and more intuitive way to edit facial expressions. In detail, we use the same tool (GAN, for instance) twice: first, to generate experimental stimuli for the reverse correlation procedure, and then to apply the resulting prototype for the manipulation. This can ensure that the manipulation is consistent with the mental representation of the observer.

Finally, the mental prototype extracted in the intermediate step (i.e., reverse correlation procedure) does not especially fit any so-called universal prototype. It is specific to the observer. To enhance the definition of facial expression prototypes, we introduce below the concept of dominant and complementary action units to precisely describe facial expression prototypes.

## 1.3 Related work

Here, we briefly review the facial expression manipulation techniques (FEM) from computer science, and the reverse correlation procedure from cognitive science.

**1.3.1 Facial expression manipulation.** Generative Adversarial Networks (GANs) [1, 2, 15] have achieved a series of impressive results in image-to-image translation tasks. This technique has widely spread to diverse domains such as art [19], medical research [20], and entertainment [21, 22]. The inputs of the model are usually a face image and a set of control parameters. The output is a new face image. According to the control parameters for manipulation, the manipulation can be divided into two categories: global attribute manipulation and local attribute manipulation.

*Preamble.* In detail, global attribute manipulation involves altering the overall appearance of the face. These attributes encompass aspects such as gender, age, face shape, face structure [6, 23, 24], and emotion [4, 25, 26]. On the other hand, local attributes are specific features or localized regions that can be individually modified to alter the facial appearance. These attributes focus on fine-grained details and localized features, such as hairstyle, presence of facial hair (beard, mustache), presence of accessories (glasses, earrings) [6, 23, 24], specific facial components (e.g., mouth shape [27–29], cheek, and eyebrows [29]) and facial muscles [3, 18]. These attributes usually do not alter the entire facial structure and identity.

As one of the representative models, the StyleGAN family (StyleGAN [6], StyleGAN2 [23], StyleGAN3 [24]) provided state-of-the-art architectures to generate high-resolution and more realistic faces. These models were able to manipulate not only global attributes such as gender, age, skin, face shape, and bone structure but also local attributes such as hairstyle, beard, mustache, and wearing accessories. Moreover, some StyleGAN-based models achieved editing photo-related features (global attributes) such as rotating, lighting, and super-resolution [21, 22].

However, all of these mentioned approaches [6, 21–24] are **face-based**. They do not contribute to facial expression manipulation (FEM) since these approaches edit neither the global attributes such as emotions nor the local attributes such as specific facial components (eyebrows, eyelids, nose, etc.) and facial muscles. On the premise of FEM, we focus on the global attribute manipulation that edits the overall expressions such as emotions, and the local attribute manipulation that edits specific facial components and facial muscles.

*Global attribute manipulation.* As a representative FEM task, face reenactment mainly aims at animating facial expressions from the source video to the target video [30–34]. As a result, the entire facial expressions (global attributes) are copied from the source video to the target video. The other attributes that are not related to the facial expression such as gender, age, face shape, face structure (global attributes), and hairstyle, beard, accessories (local attributes) are totally the same as the target video. In this task, the Face2Face model [31] and the dual-generator-based approach [32] use 3D landmarks to encode head pose, face shape, and facial expressions, and approaches such as ReenactGAN [33] and FReeNet [34] use 2D facial landmarks.

Differing from face reenactment that directly copies facial expressions, the following approaches aim at editing the overall expressions. StarGAN [4] proposed a unified model that transfers real faces from one of the six basic emotions to another. G2-GAN [25] employed the overall facial geometry as controllable parameters to synthesize the six basic emotions. By modeling the motion of facial landmarks as curves encoded as points on a hypersphere, Otberdout et al. [26] proposed an approach that generated the six basic emotions from a given neutral face.

Although the purpose of these mentioned approaches is generating realistic facial expressions, manipulating global attributes for FEM can not fulfill the aforementioned requirements (see Subsection Requirements). **Flexibility**: For a given emotion, the prototypes of this emotion can be limited by the database. For instance, if all the happy faces in the database have lip corners raised and mouths opened, the model cannot generate a happy face with lip corners raised but with mouth closed. **Exhaustiveness**: Due to the limitation of the database that only a few emotion labels are available, these FEM models [4, 25, 26] that directly controlled global attributes (such as emotion labels and overall facial landmarks) can only deal with limited emotion categories such as the six basic emotions of Ekman. However, the reality is that there are already more than 4000 emotional labels [12].

*Local attribute manipulation.* Otberdout et al [29] proposed an approach both modeling the temporal dynamics of expressions and deforming the neutral mesh to obtain the expressive counterpart. This approach achieved dynamically generating different local facial movements mostly around the mouth (e.g., bare teeth, high smile, lips up, mouth down, mouth extreme,

mouth open). Zaied et al [27, 28] proposed geometric and geometric-machine learning methods mainly to personalize smiles. However, these approaches [27–29] mainly manipulate the region around the mouth. Departing from these approaches, GANimation [3] can generate anatomically-aware face animation by taking a list of action units (AUs) as input. AUs [13], proposed by the psychologist Paul Ekman, are defined by the contraction or relaxation of one or some muscles. Differing from the emotion labels (i.e., global attributes) that require human interpretation [11], these local attributes are objective. They only contain local anatomical information about the face and can be used in combination to describe facial expressions. For instance, the prototype of happiness (proposed by Ekman) [13] is the combination of AU6 (cheek raiser) and AU12 (lip corner puller). These local attributes (i.e., AUs) involve vast facial areas such as eyebrows, eyelids, cheek, nose, mouth, and jaw. For more information, see S1 Appendix in supporting information.

In this paper, we chose GANimation [3] as a tool to synthesize facial expressions by controlling local attributes (i.e., AUs). Compared to editing global attributes such as emotions of the face, editing AUs (i.e., local attributes) can achieve fine-grained control and thus have the potential to personalize facial expressions that meet different needs of different observers. This has the potential to fulfill the requirement of **Flexibility**. Even though GANimation [3] can generate various facial expressions by combining different AUs, it lacks the ability to determine which AUs should be activated to generate the desired facial expressions that are not explicitly available in the existing database. This limitation arises from the need for expert knowledge to identify the appropriate AUs for generating the desired facial expression. This cannot meet the requirements of **Exhaustiveness** and **Expertise-free**. To the best of our knowledge, GANimation is the appropriate tool for the moment, but it can be replaced in the future by other state-of-the-art tools that manipulate local attributes.

**1.3.2 Reverse correlation.** The reverse correlation process is a powerful data-driven method widely used in the field of cognitive science to extract mental representations of observers (or called participants). Based on observers' judgments of a large quantity of randomly-varied stimuli, reverse correlation is able to reverse-engineer what perceptual representations subtend these judgments [17]. This can help researchers to identify the neural mechanisms and processing strategies involved in perception. This process is widely employed to study the perception of faces [10, 18, 35, 36], speech [37–39] and bodies [40, 41]. Note that the works [40, 41] use reverse correlation to understand how humans identify gender via bodies, and the work [35] focuses more on identity, gender, with/without expression via faces. These works are far from the research on facial expressions.

In an influential example, Jack et al. [10] randomly generated 4800 trials (i.e., stimuli generation). Each trial consists of one dynamic facial animation (called stimuli) created by the 3D morphing tool of [18]. 15 Western Caucasian and 15 East Asian observers were asked to categorize the random animations into six basic emotion categories (i.e., perceptual experiment). The authors then used reverse correlation to extract one mental representation of each emotion for each cultural group ((i.e., mental representation computation)) and conclude that these representations were, in fact, not culturally universal. This work can in principle produce control parameters for its generative model, i.e., generate a 3D synthetic face that maximizes the probability that a given observer judges it representative of one of the tested emotion categories.

As aforementioned, we choose GANimation [3] as a tool to edit AUs (local attributes) thus flexibly altering facial expressions. This can fulfill the first requirements (see Requirements), yet the second and the third requirement (i.e., **Exhaustiveness and Expertise-free**) cannot be addressed. In order to fulfill all requirements, we can use reverse correlation process to obtain the mental prototype exclusively for the observer. According to the mental prototype, the

system can identify which AUs need to be activated to generate facial expressions exclusively for the observer (i.e., personalized). The entire reverse correlation process does not rely on expert knowledge (such as FACS [13], psychology) since reverse correlation requires only the observer's perception (i.e., subjective judgment on the stimuli).

To sum up, we propose a novel interdisciplinary approach by combining the reverse correlation process with GANimation to fulfill all the aforementioned requirements: **Flexibility, Exhaustiveness, and Expertise-free**. As shown in Fig 1, GANimation can randomly generate arbitrary stimuli by editing different AUs (local attributes). By adopting reverse correlation process, the mental prototype of the observer (a vector of AUs) can be extracted. This prototype can be regarded as the control parameter of GANimation and finally, GANimation synthesizes the personalized facial expression of the observer.

## 2 Method

Our approach is composed of four successive steps (in Fig 1). In the first step (Stimuli generation), based on the real face of an actor, a generative model (denoted by GAN) is applied to synthesize a large number of arbitrary facial expressions (i.e., reverse-correlation stimuli). Then (in Perceptual experiment), the observer performs a perceptual experiment of reverse correlation in which the input is the generated stimuli. Next (in Mental representation computation), based on the responses of the observer, we compute the dominant AU and the complementary AUs and then construct the mental representation (i.e., personalized control parameters). And (in Personalized manipulation), according to the mental representation, we employ the same generative model (i.e., GAN) to generate the personalized facial expression that meets the observer's expectation. At the end (in Experiment setting), we detail the setting in terms of implementation and experiment.

### 2.1 Stimuli generation

To generate input stimuli (random facial expressions) for reverse correlation, we can employ any tool that can control objective local attributes. Here, we choose GANimation [3] controlled by facial action units (AUs) [13] to synthesize random facial expressions (i.e., reverse-correlation stimuli). In this step, GANimation takes as input an image of the actor's face (e.g. captured with an emotionally neutral expression) and a vector of AUs to create a deformed face (i.e., stimulus).

In terms of the vector of AUs, GANimation is capable of manipulating 16 AUs by activating or deactivating the corresponding AU. While simultaneously activating too many AUs typically will create visual artifacts. Therefore, we generate stimuli by only activating 3 AUs. Combining 3 out of 16 AUs, there can be $C_{16}^3 = 560$ possible AU vectors, where $C$ is the mathematical combination function. For more details about GANimation, please see the literature [3] and S1 Appendix.

### 2.2 Perceptual experiment

The second step of our approach is the perceptual experiment. For each perceptual experiment, observers perform $m$ trials. In each trial of the perceptual experiment, a pair of randomly generated stimuli is presented to the observer. Each pair of randomly generated stimuli is displayed only once. The observer is asked to choose which stimulus of the given pair best corresponds to the target expression (e.g., "which of these two faces looks happier?"). Note that for one trial, if we randomly select a pair of 3 AU-activated stimuli, there are $C_{560}^2 \approx 1.56 \times 10^5$

possible combinations. For each perceptual experiment, the set of $m$ trials is randomly selected from the $C_{560}^2 \approx 1.56 \times 10^5$ trials.

## 2.3 Mental representation computation

Here, we obtain the mental representations by a two-step computation: dominant action unit computation and complementary action units computation. The purpose of the dominant action unit computation is to determine which action unit has a significant effect on the perception of the observer. We then determine the complementary action units, i.e., in combination with the dominant action unit, which action units also drive the perception of the observer.

We'll use mathematical notation to avoid any ambiguity. To this end, we define $\Omega$ as all the trials within a perceptual experiment, where $|\Omega| = m$. Note that in this paper, $|.|$ represents the cardinality of the set. According to the activation or deactivation of a given action unit $AU_i$ ($i$ is the subscript number of AU), each perceptual experiment $\Omega$ can be divided into three subsets.

- $\Omega_{\{i^*\}}$: the subset of trials in which one of the paired stimuli has $AU_i$ activated and another one has $AU_i$ deactivated.

- $\Omega_{\{i\}}$: the subset of trials in which both stimuli have $AU_i$ activated.

- $\Omega_{\{\bar{i}\}}$: the subset of trials in which both stimuli have $AU_i$ deactivated.

**Dominant action unit computation**. We first define $Z_{\Omega_{\{i^*\}}}$ as the set of stimuli selected by the observer from the subset $\Omega_{\{i^*\}}$, and $\Phi_i$ as the set of all stimuli in which $AU_i$ is activated. We then count $P(i|\Omega_{\{i^*\}})$ the proportion of the selected stimuli that have $AU_i$ activated in the subset $\Omega_{\{i^*\}}$, i.e., how likely an activated $AU_i$ is to drive the observer's perception.

$$P(i|\Omega_{\{i^*\}}) = \frac{|Z_{\Omega_{\{i^*\}}} \cap \Phi_i|}{|Z_{\Omega_{\{i^*\}}}|} \tag{1}$$

Finally, we can determine the action unit $AU_i$ with the largest proportion $P(i|\Omega_{\{i^*\}})$ as the dominant action unit denoted by $AU_d$. $d$ is the subscript number of dominant AU.

$$d = \arg\max_i P(i|\Omega_{\{i^*\}}) \tag{2}$$

**Complementary action units computation**. Similar to the definition of $\Omega_{\{i\}}$, we specify $\Omega_{\{d\}}$ as the subset of trials where both stimuli have dominant $AU_d$ activated. We continue to divide subset $\Omega_{\{d\}}$ into three subsets according to the activation status of the non-dominant AU (denoted by $AU_j$, where $j$ is the subscript number of AU).

- $\Omega_{\{d,j^*\}}$: under the premise that each pair of stimuli has $AU_d$ (the dominant action unit) activated, the subset of trials in which one of the paired stimuli has $AU_j$ activated and another one has $AU_j$ deactivated.

- $\Omega_{\{d,j\}}$: under the premise that each pair of stimuli has $AU_d$ (the dominant action unit) activated, the subset of trials in which each pair of stimuli has $AU_i$ activated.

- $\Omega_{\{d,\bar{j}\}}$: under the premise that each pair of stimuli has $AU_d$ (the dominant action unit) activated, the subset of trials in which each pair of stimuli has $AU_i$ deactivated.

Since previously the dominant action unit has been determined, the complementary action units can not be the dominant one (i.e., $\forall AU_j \neq AU_d$). We define $Z_{\Omega_{\{d,j^*\}}}$ as the set of stimuli

selected by the observer from the subset $\Omega_{\{d,j^*\}}$, and $\Phi_j$ as the set of all stimuli in which $AU_j$ is activated. We compute $P(j|\Omega_{\{d,j^*\}})$ the proportion of selected stimuli in subset $\Omega_{\{d,j^*\}}$ that have $AU_j$ activated, i.e. how likely the addition of $AU_j$ to dominant $AU_d$ is to drive the observer's perception.

$$P(j|\Omega_{\{d,j^*\}}) = \frac{|Z_{\Omega_{\{d,j^*\}}} \cap \Phi_j|}{|Z_{\Omega_{\{d,j^*\}}}|} \tag{3}$$

In practice, we limit the number of complementary action units by introducing a threshold $T_q$ (to separate complementary AUs and non-complementary AUs).

$$\mathcal{C} = \{j|P(j|\Omega_{\{d,j^*\}}) \geq T_q\} \tag{4}$$

Note that $\mathcal{C}$ is the set of all the subscript numbers of complementary AUs.

The output of this step is the mental representation (i.e., personalized control parameters for facial expression manipulation). We construct the mental representation of the observer (as aforementioned in Stimuli generation, a 16-dimensional binary AUs) by activating the dominant action unit and all the complementary action units.

## 2.4 Personalized manipulation

Once the mental representation of the observer is extracted, we apply personalized manipulation on each frame. To be consistent with the mental representation and the final manipulation, we employ the same tool (GANimation [3]) for the stimuli generation and the personalized manipulation. To make the video compatible with GANimation (especially with the dimension of the face), we crop, align and resize the face of the actor in each frame.

## 2.5 Experiment setting

**Implementation: GANimation model**. We choose GANimation [3] as the tool to generate facial expressions by editing local attributes, namely action units (AUs) [13]. We use the code of GANimation released by its authors. All settings are unchanged. The input image and the output image are $148px \times 148px$. To crop, align, and resize the face, we employ OpenFace [42].

**Implementation: Mental representation computation**. As aforementioned, we determine a dominant AU for one emotion as the action unit that dominantly drives the observer's perception. For the complementary AUs, we need to determine which AUs combined with the dominant AU have a significant effect on driving the observer's perception. Therefore, we need to set a relatively high threshold $T_q$ to eliminate most AUs with less significant proportions $P(j|\Omega_{\{d,j^*\}})$. Indeed, $T_q = 50\%$ corresponds to the situation in which, among each pair of $AU_d$-activated stimuli, the observer selects as many $AU_j$-activated stimuli as $AU_j$-deactivated stimuli. This means that $AU_j$ carries no information content for this experimental task. To identify which AU is activated that positively influences the perception of the observer, the threshold $T_q$ should be significantly higher than 50%. Considering that state-of-the-art prototypes [13, 18] have 2 to 5 AUs activated, to align with state-of-the-art prototypes, we manually set the threshold $T_q$ to 80% for happiness, sadness and anger and 70% for self-confidence. Thus all the personalized prototypes can have 2 to 5 AUs activated.

**Experimental protocol: Observers**. Four observers (one female) participated (in October 2021) in the perceptual experiment, all relatively young (mean = 27.7yo) adults of three cultural groups: Brazil (1), China (2), and France (1), respectively denoted by observers #1 to #4. Only one observer had experience in affective computing, and nobody is a certified coder in

Facial Action Coding System [13] or a psychologist. Each observer signed informed consent, and the experimental data were anonymous.

**Experimental protocol: Perceptual experiment**. The perceptual experiment aims to illustrate that our approach can personalize the facial expressions of a given emotion, even though this emotion is not available in existing deep-learning databases. Note that the purpose of the perceptual experiment is not to give extensive results or to discuss facial expression prototypes. We chose three basic emotions (happiness, sadness, and anger) that existed in deep-learning databases and one non-basic emotion (self-confidence) that is not explicitly available in existing deep-learning databases. Each of the four observers participated in four different experimental tasks to extract his/her mental representation of happiness, sadness, anger, and self-confidence. In related work using reverse correlation [10, 36, 37, 39], the average number of trials for the perceptual experiment of one emotion varies from 700 to 1800. For each experimental task, we decided that observers performed $m$ = 840 trials. The question was fixed and unique, e.g., "Which of these two faces looks happier?" The order of the four experimental tasks was counterbalanced among observers, and all experimental tasks used the same actor's photograph. It took about 40 to 60 minutes for one observer to complete a task. The time interval between experimental tasks was set to half a day. All experiments were conducted in a quiet room in the lab, using a custom computer graphic interface implemented in PsychoPy.

**Ethics statement**. Our work does not require an ethics statement, since the risk in our work is minimal. 1) The identity of individuals is not known. 2) There is no way to track them from the data in the database. 3) There is no social or physical risk. 4) The psychological risk is absent since we asked participants according to their perceptions. 5) All the observers and participants signed informed consent forms and all the data were totally anonymous.

## 3 Results and discussion

For the results, we adopt an example from an observer to illustrate and discuss the dominant and complementary AUs computation in Dominant and complementary AUs computation, then list and discuss all personalized prototypes and the corresponding manipulations in Personalized prototypes.

### 3.1 Dominant and complementary AUs computation

Fig 2 details dominant and complementary AUs computations of each emotion (happy, sad, angry, and confident) from observer #2. See S1–S3 Figs for the computations from other observers. As mentioned in Mental representation computation, the proportion of each AU is computed based on the corresponding subset of trials (see Eqs (1) and (3)).

The concept of dominant and complementary AUs contains more information about emotional prototypes than a list of activated AUs in the universal prototypes [13]. Here are our observations. Similar observations can be found in supporting information (S1–S3 Figs).

**The dominant AU drives the observer's perception**. As defined in Mental representation computation, the dominant AU is the $AU_i$ with the largest proportion $P(i|\Omega_{\{i^*\}})$, where $i \in \mu$. As shown in the first row of Fig 2, the corresponding proportions exceed 80%. This means the observer has a significant probability to choose the facial expression that has the dominant AU activated.

**We can observe the dependency between the dominant AU and the complementary AUs**. For the dominant AU computation shown on the first row of Fig 2, the complementary AUs have much lower proportions than that of the dominant AU. For instance, in the histogram for dominant AU computation of "happy", the proportion of AU6 being selected is just 52% (note that AU6 is later determined to be the complementary AU). That is to say in the

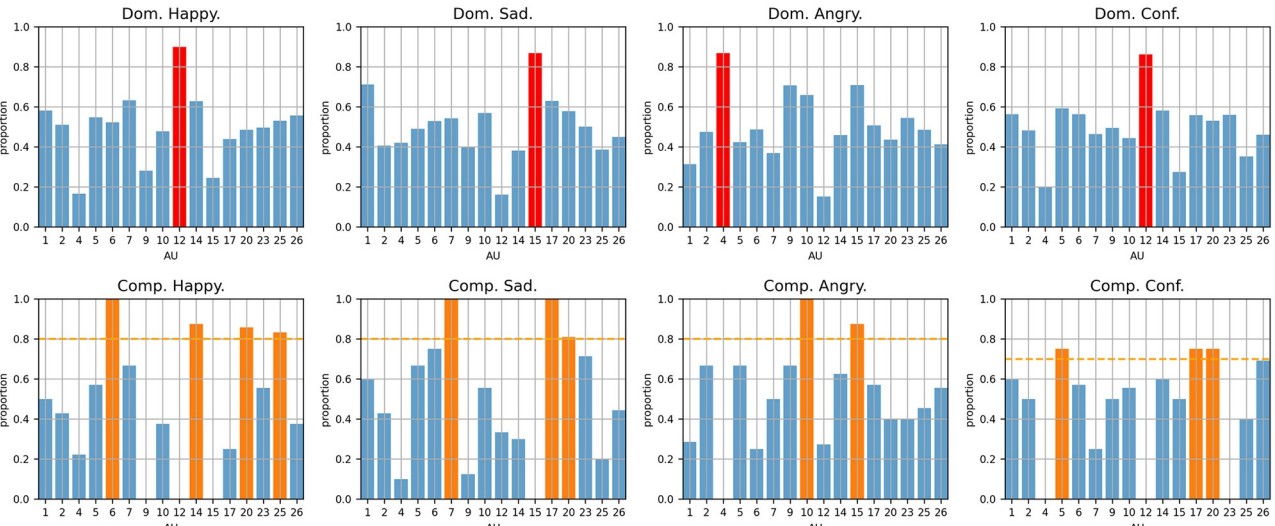

**Fig 2. The results for dominant and complementary AUs computation from observer #2.** Note that "Dom." = dominant AU computation; "Comp." = complementary AUs computation; "Conf." = self-confident. In each chart, the proportion for each AU is computed based on the corresponding subset of trials. We highlight the dominant AU in red and the complementary AUs in yellow. The thresholds for the complementary AUs computation are marked by yellow dashed lines.

following situation when the observer saw a pair of stimuli that one had AU6 activated and another one did not have AU6 activated; the observer #2 made a nearly random selection (i.e., 52% to select the stimulus that had AU6 activated and 48% to select the stimulus that did not have AU6 deactivated). This indicates that a single complementary AU can not drive the observer's perception as much as the dominant AU. However, for the complementary AU computation shown on the second row of Fig 2, when the dominant AU is activated, the facial expressions that have the complementary AUs activated have a significant probability of being selected by the observer. For instance, in the histogram for complementary AUs computation of "happy", the proportion of AU6 being selected is 100%. That is to say in the following situation when the observer saw a pair of stimuli, both stimuli had AU12 activated, one of them had AU6 activated but another one did not have AU6 activated; the observer #2 always chose the stimulus that have AU6 activated. This means that complementary AUs can drive the observer's perception only in combination with the dominant AU.

## 3.2 Convergence efficiency

We discuss here the convergence efficiency of our approach by monitoring the convergence of 1) dominant AU computation and 2) complementary AUs computation as we increase the number of trials used in the reverse correlation procedure.

To do so, we compute 1) the correlation between the histogram for dominant AU computation using the first $n$ trials from the entire perceptual experiment (i.e., $\Omega$), and the final histogram of dominant AU computation (in Fig 2); 2) the correlation between the histogram of complementary AUs computation using the first $n$ trials from the subset of trials in which all the stimuli have the dominant AU activated (i.e., $\Omega_{\{d\}}$), and the final histogram of complementary AUs computation (in Fig 2).

Fig 3 shows the convergence of dominant AU computation and complementary AUs computation from the perceptual experiment of confidence. Similar converging curves from the

## a) Dominant

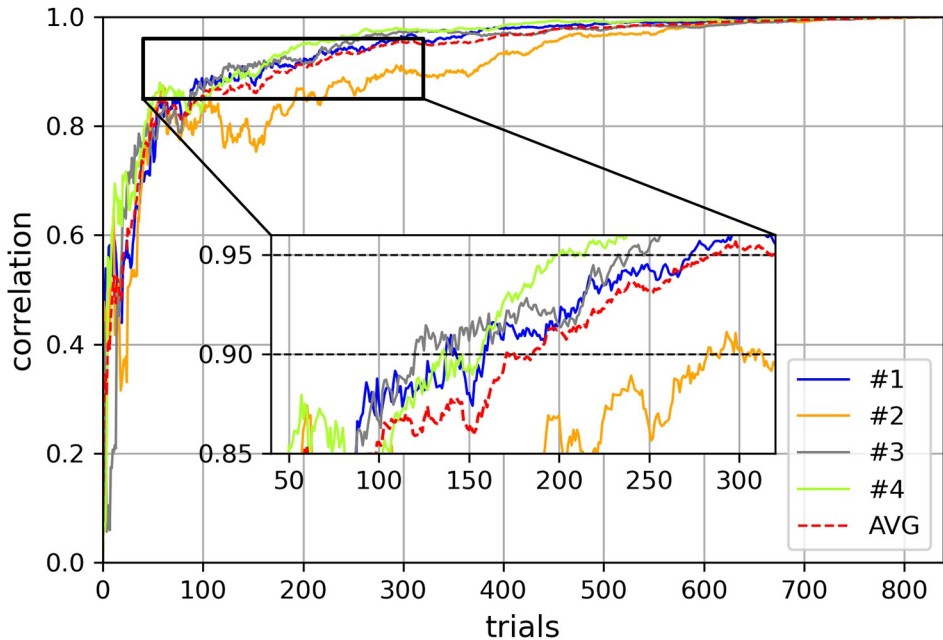

## b) Complementary

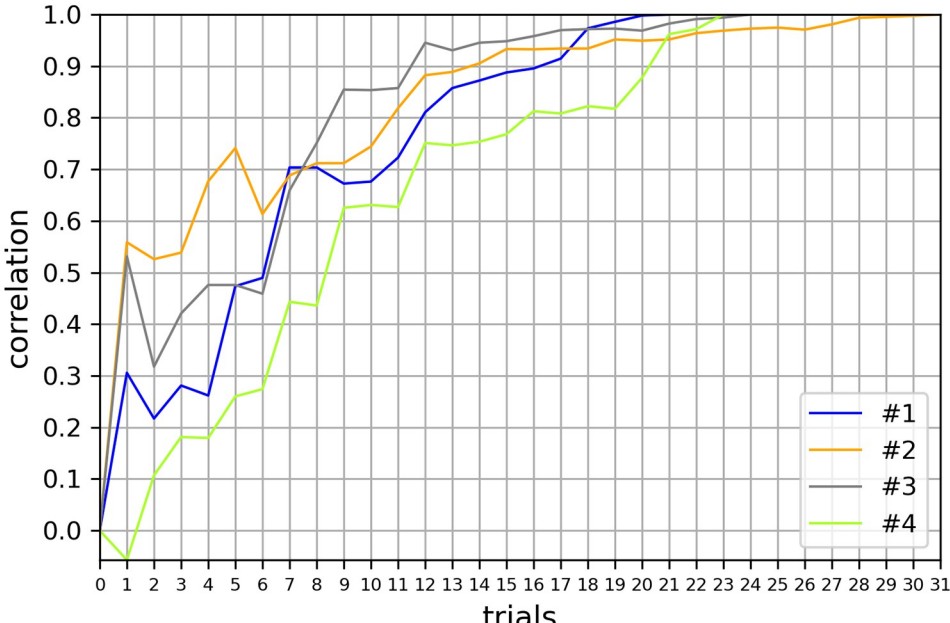

**Fig 3. Example from the perceptual experiment of confidence to monitor the convergence of our approach.** (a) Correlation between the result of dominant AU computation after the first $n$ trials (x-axis) and the result after 840 trials. The average correlation for all observers is marked by the red dashed line. (b) Correlation between the result of complementary AUs computation after the first $n$ trials (x-axis) in the corresponding subset and the result using all trials in the corresponding subset.

perceptual experiments of happiness, sadness, and anger can be found in S4–S6 Figs. These curves reflect the typical reverse-correlation convergence (see e.g., Figure 6 in [39]).

**The dominant AU can be determined with only a 12-minute experiment**. For the convergence of the dominant AU computation (in Fig 3a), while all 840 trials are considered, it takes less than 170 trials to reach a correlation of 0.9. As our approach just takes the AU with maximum proportion to determine the dominant AU (see Eq 2), performing 170 trials is enough. That is to say, only $170/840 \approx 20\%$ of the trials are necessary (equivalent to 12 minutes if the entire perceptual experiment needs 60 minutes).

**Only a few data are used for complementary AUs computation**. For each observer, only a small subset of trials (i.e., $\Omega_{\{d\}}$) are, in effect, used to estimate complementary AUs. For instance, Fig 3b illustrates that the largest subset, which is from observer #2, only includes 31 trials, and these trials are distributed throughout the entire perceptual experiment.

**A prototype can be determined accurately in about 20 minutes**. Although in our approach, the number of trials is set based on related works [10, 36, 37, 39], it appears unnecessary to randomly generate as many as 840 trials to determine dominant and complementary AUs for an observer. In fact, the duration of the perceptual experiment can be largely reduced. If our approach only randomly generates the first 170 trials to determine the dominant AU and then generates another 100 trials only from $\Omega_{\{d\}}$ to determine the complementary AUs, it will be less than 20 minutes (270 trials, instead of 840 trials) to obtain the mental prototype.

## 3.3 Personalized prototypes

We reconstruct the personalized prototypes of each observer by activating the dominant and complementary AUs. Fig 4 shows the personalized prototype of each observer, as well as state-of-the-art prototypes from the literature [13, 18] (denoted by "Ek" and "Yu"). The corresponding activated AUs are listed at the bottom of the faces. For the personalized prototypes, we highlight the dominant AU in square brackets; the others are the complementary AUs. Note that there is not any published database of confidence, and there is no state-of-the-art prototype for confidence.

According to the results in Fig 4, we observe that **the personalized prototypes are compatible with state-of-art prototypes**. All 12 prototypes of basic emotions generally convey expressions similar to that of Ekman [13] and Yu [18]. All dominant AUs: AU12 for happiness, AU4 or AU15 for sadness, and AU4 or AU9 for anger, can be found in state-of-the-art prototypes.

**The prototypes are personalized**. In each perceptual experiment task, although observers were asked the same questions, all observers acquired subtly different mental representations and, especially, different complementary AUs. With the exception of observer #1-sad and Ekman [13], all synthesized facial expressions also differed from the state-of-the-art prototypes by at least one AU. For instance, observer #2-happy is the same as Yu [18], plus the addition of AU20 (lip stretcher), resulting in a wider smile.

**Our manipulations can be extended to the emotions that are not available in existing databases, such as confidence**. We had no comparison prototypes for confidence. Although all confidence manipulations had the same dominant AU12 as happiness, the expressions remained different from any of the listed prototypes of happiness, notably because of the involvement of AU4 (brow lowerer), AU5 (upper lid raiser), AU9 (nose wrinkler) and AU17 (chin raiser).

## 4 Evaluations

Here, we evaluate the personalized prototypes to prove the effectiveness of our approach. In Subjective evaluation by mean opinion score, we conduct a subjective evaluation with the

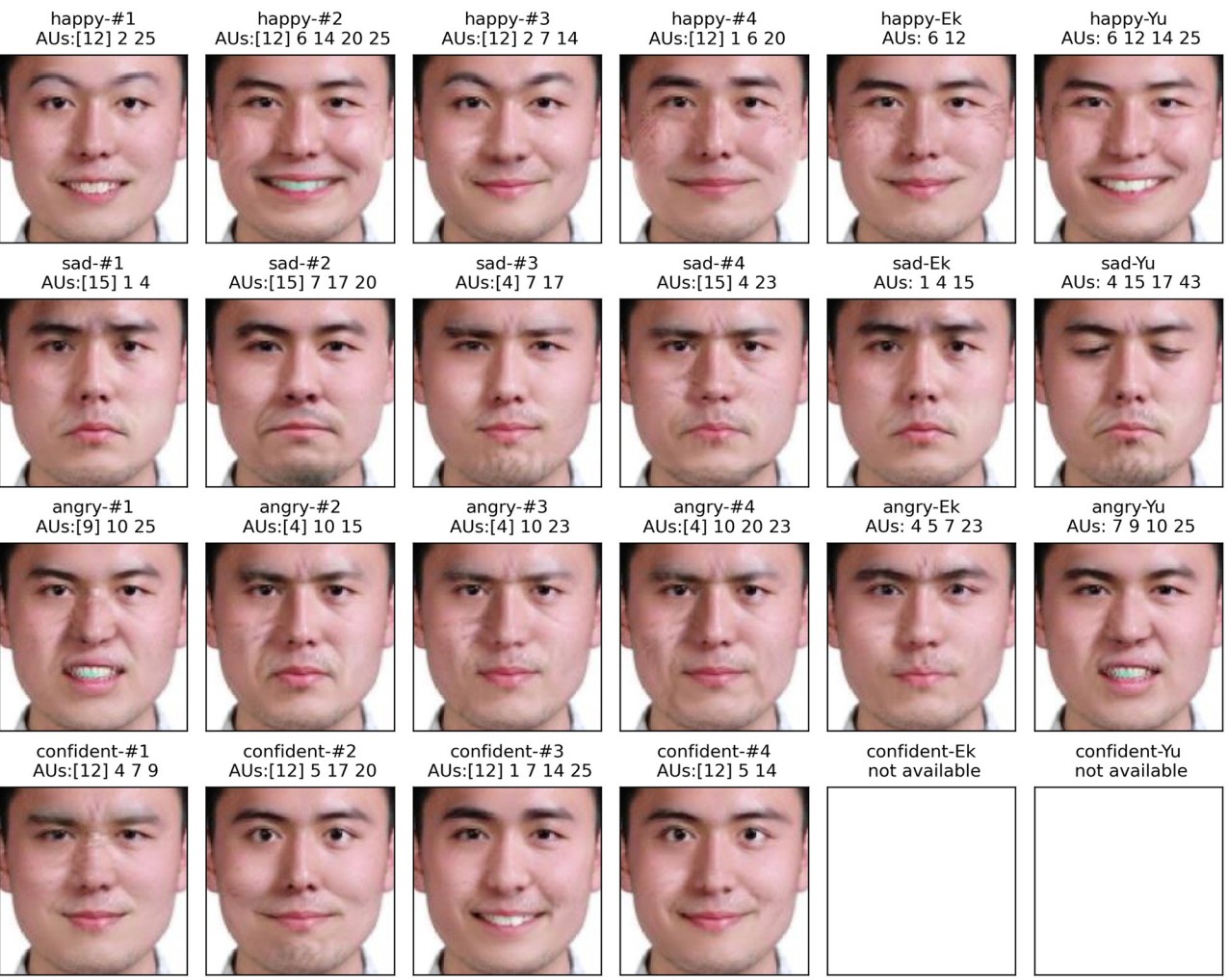

**Fig 4. Personalized prototypes of the four emotions from observers and state-of-the-art prototypes [13, 18].** For each personalized prototype, we indicate the emotion category followed by the observer (denoted by "#1" to "#4") or the state of the art (denoted by "Ek" and "Yu"). We detail the dominant AU in square brackets and the others are complementary AUs. For the state-of-the-art prototypes, we only list the activated AUs. All facial expressions are reconstructed by the same GAN (i.e., GANimation [3]) and the same actor. Note that GANimation [3] cannot edit AU16 (lower lip depressor). We replace AU16 with AU25 (lips part) to reconstruct the angry prototype of Yu [18] ("angry-Yu").

people who participated in the perceptual experiment, i.e., observers. The purpose of this evaluation is to assess the satisfaction of each observer with the prototypes they have created. In Subjective evaluation by ranking, we conduct another subjective evaluation by two diverse groups of people: observers and non-observers. We aim to quantify the acceptance of the personalized prototypes and compare them with the state-of-art prototypes [13, 18]. Note that the purpose of this paper is not the extensive discussion of prototypes, such as their impact on affective states across different cultures, but only to validate the effectiveness of the procedure.

## 4.1 Subjective evaluation by mean opinion score

To assess what observers thought of their own personalized facial expression prototypes, we employed the Mean Opinion Score, which is a popular indicator of perceived media quality [43]. We asked the four observers to rate their personalized facial prototypes from 1 to 5

**Table 1. Subjective evaluation by mean opinion score.**

| emo.＼obs. | #1 | #2 | #3 | #4 | mean |
|---|---|---|---|---|---|
| Happy | 4 | 5 | 5 | 4 | 4.5 |
| Sad | 5 | 5 | 4 | 4 | 4.5 |
| Angry | 5 | 4 | 4 | 4 | 4.25 |
| Conf. | 4 | 5 | 4 | 5 | 4.5 |

Each observer (denoted by #1, #2, #3, and #4) rated their personalized prototypes. The mean opinion score for each emotion (denoted by emo.) is shown in the last column. Note that Conf. = self-confident.

representing bad satisfaction to excellent satisfaction. In Table 1, we listed all the scores rated by observers and presented the mean opinion score for each emotion (happy, sad, angry and confident) in the last column.

All observers rated their personalized facial prototypes with "4" and "5", meaning that **observers were satisfied with their personalized facial prototypes**.

## 4.2 Subjective evaluation by ranking

To quantify the acceptance of the personalized prototypes, we added the state-of-art prototypes of Ekman and Yu [13, 18] for comparison. We conducted a ranking study involving two distinct groups of people: 1) The "observers" group consisted of the observers who had previously performed Perceptual experiment (i.e., observers #1, #2, #3, and #4), and 2) The "non-observers" group comprised participants recruited from Amazon Mechanical Turk (AMT) who had not performed the previous Perceptual experiment.

Based on the ranking results obtained from the "observers" group, we aimed to investigate whether each observer still preferred his/her own personalized prototype when presented with a set of prototypes mixed with other observers' personalized prototypes and state-of-the-art prototypes. Additionally, by analyzing the ranking results obtained from the "non-observers" group, we aimed to gain insights into the preferences of the other people (bystanders) who did not participate in the previous perceptual experiments.

**4.2.1 Procedure.** In detail, each one performed 4 ranking tasks. Each task corresponds to one of the four emotions. In each ranking task, all prototypes (6 listed prototypes for happy, sad, and angry, 4 listed prototypes for confident as in Fig 4) were presented anonymously and shuffled. Participants were asked to rank these faces from the happiest / saddest / angriest / most confident to the least. Everyone was informed that all data collected were totally anonymous.

**For the "observers" group**. In order to investigate if the prototypes are really personalized, we calculate the probability that each observer ranks his/her own personalized prototypes in each position (i.e., from the 1st place to the 6th place).

**For the "non-observers" group**. Since "non-observers" are not involved in personalizing these prototypes, the measurement applicable to the "observers" group is not suitable for the "non-observers" group. Therefore, we compared personalized prototypes as well as state-of-the-art prototypes (i.e., baseline) in a relatively objective way. To analyze the rankings from all the AMT participants, we assess the preferences of the prototypes in the following two steps. 1) We first counted for each possible pair of prototypes how many participants preferred one of the prototypes to the other. 2) We then employed the Schulze voting method [44] to compute the preferences between each pair of prototypes and to derive the final ranking of these prototypes.

Moreover, in the first step, according to the ranking, some preferences are cyclic: for sadness, 51% of the participants preferred the prototype of observer #2 to the prototype of observer #4, 57% of the participants preferred the prototype of observer #4 to the state-of-the-art prototype of "Yu" [18], whereas 52% of the participants preferred the state-of-the-art prototype of "Yu" [18] to the prototype of observer #2. Thus, we can not directly quantify the acceptance between these prototypes. That's the reason why we employ the Schulze method to compute the preferences in the second step. For more details about the Schulze voting method, please see the literature [44] and the supporting information (S2 Appendix).

**4.2.2 Results.** In terms of the "observers" group, Table 2 illustrates that each observer ranks his/her personalized prototypes in each position. Note that there are only 4 prototypes of self-confidence (no state-of-the-art prototype of self-confidence), whereas, for the other emotions, there are 6 prototypes. That is why the probability of "5th" and "6th" by random ranking is different from the others.

We notice that all the observers rank their personalized prototypes in the top-3. All the observers have at least a 50% probability to rank their personalized prototypes in the first place. However, if the ranking is random, the probability to select their personalized prototypes is only 18.75%. **This indicates that the prototypes are personalized. They can reflect the observer's mental image and well answer the question in the perceptual experiment**.

In terms of the **"non-observers" group**, in Table 3, we present the preferences between each pair of prototypes computed by the Schulze method and the final rankings. Note that due to cyclic preferences (such as the aforementioned "#2", "#4", and "Yu." of sadness) the sum of the paired preferences is not always equal to 100%. For instance, in Table 3(b), the sum of the preference from "4" to "Yu." (57%) and the preference from "Yu." to "4" (51%) is 108%. Considering state-of-the-art prototypes as the baselines and for these 217 participants, our observations are as follows.

- **The low-ranking personalized prototypes are about equally preferred to at least one of the state-of-the-art prototypes**. "#3", "#4", and "Ek." in Table 3(a) and 3(c), and "#2", "#3", and "Yu." in Table 3(b) are low-ranking (ranked in the last three). The paired preferences between them are around 50%. For instance, in Table 3(a), 46% of the participants preferred "4" to "Ek.", and 54% of the participants preferred "Ek." to "4". That is to say, these low-ranking personalized prototypes are about equally preferred to the state-of-the-art prototype ("Ek." or "Yu."). This also validates that our approach can generate personalized prototypes.

- **Emotional prototypes are not universal**. As shown in Table 2, the prototypes are not universally preferred among participants. Although in Table 3(a) and 3(c), the top-ranking prototypes are much preferred over the others ("#2" of happiness and "#1" of anger), most preferences are far from 100% (and 0%). Especially in Table 3(b), most prototypes of sadness

**Table 2. The probability that each observer ranks his/her own personalized prototype in each position.**

| obs. \ pos. | 1st | 2nd | 3rd | 4th | 5th | 6th |
|---|---|---|---|---|---|---|
| #1 | 75% | 25% | | | | |
| #2 | 75% | 25% | | | | |
| #3 | 50% | 50% | | | | |
| #4 | 50% | 25% | 25% | | | |
| random | 18.75% | 18.75% | 18.75% | 18.75% | 12.5% | 12.5% |

We set the probability to select their personalized prototypes by random ranking as the baseline. "obs." = observer; "pos." = position.

**Table 3. Preferences between each pair of prototypes computed by the Schulze method and the final rankings.**

| (a) happiness | | | | | | | |
|---|---|---|---|---|---|---|---|
| From \ To | #1 | #2 | #3 | #4 | Ek. | Yu. | ranking |
| #1 | - | 16% | **75%** | **70%** | **82%** | **63%** | 2 |
| #2 | **84%** | - | **82%** | **87%** | **94%** | **87%** | 1 |
| #3 | 25% | 18% | - | **54%** | **52%** | 24% | 4 |
| #4 | 30% | 13% | 46% | - | 46% | 25% | 6 |
| Ek. | 18% | 6% | 48% | **54%** | - | 15% | 5 |
| Yu. | 37% | 13% | **76%** | **75%** | **85%** | - | 3 |

| (b) sadness | | | | | | |
|---|---|---|---|---|---|---|
| From \ To | #2 | #3 | #4 | #1/Ek. | Yu. | ranking |
| #2 | - | **52%** | 51% | 48% | 51% | 4 |
| #3 | 48% | - | 42% | 38% | 46% | 5 |
| #4 | **52%** | **58%** | - | 49% | **57%** | 2 |
| #1/Ek. | **52%** | **62%** | **51%** | - | **64%** | 1 |
| Yu. | **52%** | **54%** | 51% | 36% | - | 3 |

| (c) anger | | | | | | | |
|---|---|---|---|---|---|---|---|
| From \ To | #1 | #2 | #3 | #4 | Ek. | Yu. | ranking |
| #1 | - | **70%** | **76%** | **78%** | **76%** | **84%** | 1 |
| #2 | 30% | - | **78%** | **72%** | **73%** | 45% | 3 |
| #3 | 24% | 22% | - | **52%** | **51%** | 33% | 4 |
| #4 | 22% | 28% | 48% | - | **54%** | 36% | 5 |
| Ek. | 24% | 27% | 49% | 46% | - | 34% | 6 |
| Yu. | 16% | **55%** | **67%** | **64%** | **66%** | - | 2 |

| (d) self-confidence | | | | | |
|---|---|---|---|---|---|
| From \ To | #1 | #2 | #3 | #4 | ranking |
| #1 | - | 46% | 38% | 46% | 4 |
| #2 | **54%** | - | 26% | 44% | 3 |
| #3 | **62%** | **74%** | - | **76%** | 1 |
| #4 | **54%** | **56%** | 24% | - | 2 |

The personalized prototypes of the corresponding observers are denoted by "#1" to "#4". "Ek." and "Yu." refer to state-of-the-art prototypes [13, 18]. Since the sadness prototypes of observer #1 and "Ek." are identical, we merged their preference data and denoted them by "#1/Ek.". For each pair of prototypes, we highlight the larger preferences in bold. For instance, for happiness, 84% of the participants preferred "#2" to "#1", whereas 16% of the participants preferred "#1" to "#2".

(including state-of-the-art prototypes) are about equally preferred among the hired participants. Indeed, most preferences are close to 50% which is quite far from 100%. Even there are cyclic preferences. Hence, there can be many prototypes of one emotion.

To sum up, our approach generated personalized prototypes which both differed from each other and from state-of-the-art prototypes. According to rankings from the "observers" group, we can validate that the prototypes are personalized. According to the rankings from the "non-observers" group, personalized prototypes are either close to state-of-the-art prototypes (e.g. "#1" and "Ek." of sadness are identical) or preferred to state-of-the-art prototypes. Rankings also suggest that preferred prototypes of any single emotion are not unique across different people.

## 5 Limitations and future work

The first limitation of our approach comes from the tool (GAN or any other type of local attribute manipulation tool) for personalizing prototypes. The choice of the tool can limit the number and the type of local attributes that could be manipulated. For instance, GANimation only focuses on AUs and does not consider other attributes, such as gaze direction [45], yet such attributes should ideally be integrated into the reverse correlation process. Additionally, GANimation is also incapable of editing AU16 (lower lip depressor). Although it is not the goal of this paper, the authenticity of the face textures can therefore be improved.

Another limitation of our approach comes from the reverse correlation process. Performing 840 trials (about 40 to 60 minutes) for the perceptual experiment is time-consuming. As mentioned in Convergence efficiency, the procedure can be greatly reduced, to about 270 trials, without losing accuracy. While this amount of 'training' data is several orders of magnitude smaller than what would be needed to e.g. train a GAN from a dataset of annotated examples for each emotion, the time burden on observers (about 20 minutes) can still be high in certain application contexts. An automatic optimization process can be considered to further speed up the process.

Finally, a limitation common to our approach and relative reverse correlation work is that all the stimuli are unimodal. Reverse-correlation multimodal prototypes (i.e., both how a face should look and how it should sound) have the potential to enrich affective computing studies such as [46] in the future.

## 6 Conclusion

In this paper, we proposed a novel interdisciplinary approach to personalize facial expressions by combining the facial expression manipulation technique from computer science with reverse correlation, a procedure from cognitive science able to extract personalized mental representations based on observers' judgments. Our approach can personalize manipulations of facial expressions that are not limited to basic emotions, and without the need for expertise.

We hope our approach can pave the way for further scientific studies in both affective computing and computer science, and also expect it can be customized for audiences in different application domains, e.g., a digital coach for the online interview or a digital mirror treating psychiatric disorders of emotion.

## Supporting information

**S1 Fig. Mental representation computation from the observer #1.**
(TIF)

**S2 Fig. Mental representation computation from the observer #3.**
(TIF)

**S3 Fig. Mental representation computation from the observer #4.**
(TIF)

**S4 Fig. Converging curves for happiness.**
(TIF)

**S5 Fig. Converging curves for sadness.**
(TIF)

**S6 Fig. Converging curves for anger.**
(TIF)

**S1 Appendix. GANimation [3].**
(PDF)

**S2 Appendix. Schulze method [44].**
(PDF)

## Acknowledgments

We thank the reviewers for their valuable comments.

## Author Contributions

**Conceptualization:** Sen Yan, Catherine Soladié.

**Formal analysis:** Sen Yan.

**Funding acquisition:** Renaud Seguier.

**Investigation:** Sen Yan.

**Methodology:** Sen Yan, Catherine Soladié.

**Project administration:** Catherine Soladié, Renaud Seguier.

**Resources:** Renaud Seguier.

**Software:** Sen Yan.

**Validation:** Sen Yan, Catherine Soladié, Jean-Julien Aucouturier.

**Visualization:** Sen Yan.

**Writing – original draft:** Sen Yan.

**Writing – review & editing:** Sen Yan, Catherine Soladié, Jean-Julien Aucouturier, Renaud
Seguier.

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
