## [Decision Letter · Decision Letter 0]

26 May 2023

PONE-D-23-08238Combining GAN with reverse correlation to construct personalized facial expressionsPLOS ONE

Dear Dr. Yan,

Thank you for submitting your manuscript to PLOS ONE. After careful consideration, we feel that it has merit but does not fully meet PLOS ONE’s publication criteria as it currently stands. Therefore, we invite you to submit a revised version of the manuscript that addresses the points raised during the review process.

We look forward to receiving your revised manuscript.

Kind regards,

Chenchu Xu, Ph.D

Academic Editor

PLOS ONE

“Our work was funded by Randstad group (https://www.randstad.fr/, to Sen Yan, Catherine Soladié, and Renaud Seguier), Agence Nationale de la Recherche (ANR) REFLETS (https://anr.fr/, to Catherine Soladié, Jean-Julien Aucouturier, and Renaud Seguier), and European Research Council (ERC) PoC ACTIVATE 875212 and Fondation Pour l’Audition FPA RD-2018-2 (https://erc.europa.eu/homepage, to Jean-Julien Aucouturier).”

“Our work was funded by Randstad group (to S.Y., C.S., and R.S.), Agence Nationale de 524

la Recherche (ANR) REFLETS (to C.S., JJ.A., and R.S.), and European Research 525

Council (ERC) PoC ACTIVATE 875212 and Fondation Pour l’Audition FPA RD-2018-2 526

(to JJ.A.).”

“Our work was funded by Randstad group (https://www.randstad.fr/, to Sen Yan, Catherine Soladié, and Renaud Seguier), Agence Nationale de la Recherche (ANR) REFLETS (https://anr.fr/, to Catherine Soladié, Jean-Julien Aucouturier, and Renaud Seguier), and European Research Council (ERC) PoC ACTIVATE 875212 and Fondation Pour l’Audition FPA RD-2018-2 (https://erc.europa.eu/homepage, to Jean-Julien Aucouturier).”

6. Please upload a copy of Supporting Information Figure S1 fig and S2 fig which you refer to in your text on page 13.

Additional Editor Comments:

The manuscript presents innovative research on a facial expression generation method but requires significant revisions to improve clarity, comprehension, and validation. Key recommendations include: clarifying distinctions and research in attribute manipulation, improving image quality, expanding on current FEM techniques, explaining the GANimation model in more detail, conducting a comprehensive literature survey, simplifying methodological details and notation, addressing formatting and logical flow issues, relating results to chance level, providing clearer evidence for specific claims, improving figure resolution, explaining technical terms within the manuscript, and correcting minor errors. Addressing these concerns will significantly enhance the manuscript's quality and its potential for publication.

Reviewers' comments:

Reviewer's Responses to Questions

**Comments to the Author**

1. Is the manuscript technically sound, and do the data support the conclusions?

Reviewer #1: Partly

Reviewer #2: No

2. Has the statistical analysis been performed appropriately and rigorously? 

Reviewer #1: No

Reviewer #2: No

3. Have the authors made all data underlying the findings in their manuscript fully available?

Reviewer #1: No

Reviewer #2: No

4. Is the manuscript presented in an intelligible fashion and written in standard English?

Reviewer #1: Yes

Reviewer #2: Yes

5. Review Comments to the Author

Reviewer #1: This paper presents a facial expression generation method that utilizes a generative adversarial network model combined with reverse correlation to construct personalized facial expressions. This method has significant implications in the field of facial expression manipulation. While I believe the research is important, there are still some aspects that need to be modified and improved.

-The Facial expression manipulation section would benefit from a clearer explanation of the differences between high-level attribute manipulation and low-level attribute manipulation, as well as a brief overview of existing research in these two areas.

-The quality of the images provided in this paper could be improved by regenerating high-resolution images or using a vector image format to avoid distortion caused by scaling.

-While the weaknesses of FEM techniques are well-defined and specific, more details on current efforts to address these weaknesses would be helpful. For instance, what research or development is being done to improve fine control or personalize facial expressions?

-A detailed explanation and description of the GANimation model used in the Stimuli generation section, including its advantages and limitations, as well as the reasons for choosing this model, should be provided.

-The authors should conduct a more thorough literature survey. Some relevant papers to consider include:[1] GANs and Artificial Facial Expressions in Synthetic Portraits. [2] BMAnet: Boundary Mining With Adversarial Learning for Semi-Supervised 2D Myocardial Infarction Segmentation. [3] 3D Cartoon Face Generation with Controllable Expressions from a Single GAN Image. [4] Sparse to Dense Dynamic 3D Facial Expression Generation. [5] GCFSR: a Generative and Controllable Face Super Resolution Method Without Facial and GAN Priors.

Reviewer #2: Yan et al report an experiment in which face stimuli are generated by a generative adversarial network which can be fed with binary vectors that trigger the expression of so-called action units that lead to specific emotional expressions. The authors show such stimulus material to 4 participants and ask them to rate pairs of stimuli in terms of which of the pair seems a better expression of a given emotion. The authors then conduct a somewhat convoluted analysis to derive a "mental representation" of the emotions specific to each participant, test how much of their data is needed to obtain similar results and show the reconstructions to online participants (n = 217) in order to see to what extent these online participants agree with the reconstructions.

The manuscript is written in a quite confusing way, where for example the methodological details are spread out over multiple subsections that each explain a different sub-aspect of a problem, the figure captions are mostly unintelligible, and the authors choose very quirky analyses that they describe in a maximally convoluted way which I believe will reduce the intelligibility of what was done to pretty much any audience. None of the results are related to chance level, so it is hard to infer what the authors really have "found".

I however believe that if these issues are adressed, the manuscript could be improved considerably.

Here are my concerns, in a loosely descending order of importance:

1) Logic of line 225 / "dominant action unit computation"

What is the rationale that motivated this? Why did the authors not simply consider a linear model that predicts the decision as a function of the binary vector of AUs?

Is the mathematical set notation really necessary? It is extremely tough to read, and seems like it is greatly and unnecessarily overcomplicating relatively simple matters. Line 214 / 215 seems just a bit over the top. How exactly am I to understand the final mu[k] = i? Line 232: "knowing that for all AUj != AUd" -- I have no idea what is meant here.

It seems to me that the whole notation could be summarised in a few sentences like so:

"For any given AU i, we considered the subset of trials in which only one stimulus of the pair had the AU i activated. We then divided the number of stimuli containing AU i that were chosen by the participant by the number of all trials in the subset. We defined the dominant AU as the one with the highest fraction.

We further considered trials in which both stimuli had a dominant AU activated. In a subset of these trials, a given additional AU j was active in only one of the two stimuli. We divided the number of stimuli that were chosen by the participant and had AU j active by the number of trials in this subset. For any AU j where this fraction exceeded .7 (self-confidence) or .8 (all other emotions), we counted AU j as a complementary AU."

If the authors wish to claim that the personalisation worked to some degree, then they could demonstrate this by having the evaluations run not just within participants with their own mental representations, but also with the mental representations of other participants, where they could find some difference in the evaluation? E.g. a participant could evaluate the reconstruction of another participant as lower than their own in terms of reflecting a given emotion?

2) Formating / order

I don't fully understand the figure caption formatting: Why are the captions in the main text without clear separation? How can the figure caption (or what I think is the figure caption) of figure 3 end with "Here are our observations"? That would make sense if it was part of the text, but not for a figure caption.

The caption for figure 2 is almost entirely unintelligible to me. It got clearer when finding the later line 217 - 221. The authors should also explain what they mean with dominant and complementary before relying on these terms in the figure 2 caption. What is i? What is j? What do the three rows denote? What does Omega_{i,j*} mean?

Why is the section on "convergence efficiency" listed after the mechanical turk results? It would seem more logical to me to have them in the middle between the first reverse correlation results and the validation experiment, since the same data as in the reverse correlation results are used.

Further, (but that is a subjective point and I will accept it if the authors see it differently), I feel the way the methods are noted down is somewhat confusing. Why is it necessary to first abstractly note the number of trials as m in line 197, and later specify it as 840 (line 292, in the results section -- that just seems chaotic to me). In the same way, why does the setting of the threshold (line 269) have to be separated from its description in line 237?

Table 2 Caption: The authors should explain here what they mean by "for" and "over".

3) Referencing results to chance level

Line 322: "significant" -- what is the definition here? Can the authors add a permutation-based approach that would show that exceeding 80% is indeed something that we would not expect from chance alone? The same holds for the conclusions in line 354: Are the personal components really personal to a level that exceeds chance? Or do we just see noise that looks different for each participant?

Why were the different tresholds Tq selected? Why is 51% not enough (that's more than 50%, too)? To me it seems it might make sense to derive an empirical chance level, e.g. with permutations?

4) Why was observer #2 chosen for the main figures? The main figures should include group-level results.

5) On how many trials was each dominant / complementary AU result based (i.e. how many trials were in the corresponding subsets?)

6) "corresponding complementary AUs have much lower proportions than the dominant AUs" -- I see exactly the opposite in the figure. Complementary AUs reach 1.0, which is achieved by none of the dominant AUs. What on earth do the authors refer to here?

7) Line 337: "the correpsonding AUs are listed at the bottom of the faces" -- I can't see anything there except for some undefined blobs of turquoise. The authors should consider a higher resolution of their figures.

8) Line 412: The manuscript should stand on its own. It is not enough to refer to a third source to explain the "Schulze voting method". I do not understand what the authors mean by "cyclic preference".

9) line 430: I do not see how this validates the personalised prototypes.

10) line 71: do the authors validate their claim that their model can "can cover a wide range of local facial movements"?

11) line 115: on what dimension are "high"- vs "low-level" attributes different? How are action units "low level"?

12) The authors should cite the work of Peterson et al, PNAS 2021

Typos etc:

- line 30: "artificial intelligent" -> change either to artificially intelligent (still sounds a bit weird though) or artificial intelligence

- line 51: "mental representation" -> either "the mental representation" or "mental representations"

- line 147: "was asked" -> were asked

- line 375: "not to the extensive discussion" -> "not the extensive discussion", or "not to extensively discuss"

6. PLOS authors have the option to publish the peer review history of their article (what does this mean?). If published, this will include your full peer review and any attached files.

Reviewer #1: No

Reviewer #2: No

---

## [Author Response · Author response to Decision Letter 0]

11 Jul 2023

To all reviewers and editors:

First, we would like to send my sincerest thanks to the reviewers and the editor for their efforts. 

We followed the comments of the reviewers, majorly modified the related work and the methodology and reorganized the order of the subsection “convergency efficiency”. All the supporting information were updated accordingly.

In addition, a new subjective evaluation has been added and merged with the previous subjective evaluation in section 4.2. 

In the attachment, I copied the comments from the reviewers and the editor and answered each of them accordingly.

---

## [Decision Letter · Decision Letter 1]

14 Aug 2023

Combining GAN with reverse correlation to construct personalized facial expressions

PONE-D-23-08238R1

Dear Dr. Yan,

We’re pleased to inform you that your manuscript has been judged scientifically suitable for publication and will be formally accepted for publication once it meets all outstanding technical requirements.

Kind regards,

Chenchu Xu, Ph.D

Academic Editor

PLOS ONE

Additional Editor Comments (optional):

Reviewers' comments:

Reviewer's Responses to Questions

**Comments to the Author**

1. If the authors have adequately addressed your comments raised in a previous round of review and you feel that this manuscript is now acceptable for publication, you may indicate that here to bypass the “Comments to the Author” section, enter your conflict of interest statement in the “Confidential to Editor” section, and submit your "Accept" recommendation.

Reviewer #1: All comments have been addressed

Reviewer #2: All comments have been addressed

2. Is the manuscript technically sound, and do the data support the conclusions?

Reviewer #1: Yes

Reviewer #2: Yes

3. Has the statistical analysis been performed appropriately and rigorously? 

Reviewer #1: Yes

Reviewer #2: Yes

4. Have the authors made all data underlying the findings in their manuscript fully available?

Reviewer #1: No

Reviewer #2: Yes

5. Is the manuscript presented in an intelligible fashion and written in standard English?

Reviewer #1: Yes

Reviewer #2: Yes

6. Review Comments to the Author

Reviewer #1: This manuscript is technically feasible and the research results can support the methods and conclusions. I suggest that the author make further research in the future.

I suggest that the author make the data used available to readers to support the research and arguments of others.

Reviewer #2: (No Response)

7. PLOS authors have the option to publish the peer review history of their article (what does this mean?). If published, this will include your full peer review and any attached files.

Reviewer #1: No

Reviewer #2: No

---

## [Editor Report · Acceptance letter]

17 Aug 2023

PONE-D-23-08238R1 

Combining GAN with reverse correlation to construct personalized facial expressions 

Dear Dr. Yan:

I'm pleased to inform you that your manuscript has been deemed suitable for publication in PLOS ONE. Congratulations! Your manuscript is now with our production department. 

Kind regards, 

on behalf of

Dr. Chenchu Xu 

Academic Editor

PLOS ONE